# What Is Associated with Changes in Food Security among Low-Income Residents of a Former Food Desert?

**DOI:** 10.3390/nu14245242

**Published:** 2022-12-09

**Authors:** Jonathan Cantor, Bonnie Ghosh-Dastidar, Gerald Hunter, Matthew Baird, Andrea S. Richardson, Sameer Siddiqi, Tamara Dubowitz

**Affiliations:** 1RAND Corporation, 1776 Main Street, Santa Monica, CA 90401, USA; 2RAND Corporation, 4570 Fifth Avenue, Pittsburgh, PA 15213, USA; 3RAND Corporation, 1200 South Hayes Street, Arlington, VA 22202, USA

**Keywords:** food security, nutrition, food environment

## Abstract

Lack of geographic access to foods has been postulated as a cause for food insecurity, which has been linked to poor nutrition, obesity, and chronic disease. Building on an established cohort of randomly selected households from a low-income, predominantly Black neighborhood, we examined household food security, distance to where study participants reported doing their major food shopping, and prices at stores where they shopped. Data from the Pittsburgh Hill/Homewood Research on Eating, Shopping, and Health study for years 2011, 2014 and 2018 was limited to residents of the neighborhood that began as a food desert (i.e., low access to healthy foods), but acquired a full-service supermarket in 2013. We calculated descriptive statistics and compared study participants in the former food desert neighborhood whose food security improved to those whose food security did not improve across survey waves. We estimated cross sectional linear regressions using all waves of data to assess food security level among study participants. Distance to major food shopping store was positively associated with food security (*p* < 0.05) while food-store prices were not significantly associated with food security. Findings suggest that for predominantly low-income residents, food secure individuals traveled further for their major food shopping.

## 1. Introduction

Food insecurity—a lack of consistent access to enough food for an active, healthy life—is linked to poor nutrition, obesity, and chronic disease [1,2]. Food insecurity increases hunger, malnutrition [3], and has a negative effect on health and quality of life [4]. While approximately 10.5 percent of the United States population experienced some level of food insecurity in 2019, food insecurity among Black Americans hovered closer to 19 percent [5]. Causes of food insecurity include low household income, limited food choice if residing in a community without a supermarket, and higher food prices in small stores which contributes to less affordable food within some communities [6].

Both food prices and distance to major food retail venues can be considered important predictors for food security given they impact access to food [7,8]. Several studies have examined the relationship between distance to food stores, food prices and food security [9,10]. Both studies focused on Supplemental Nutrition Assistance Program (SNAP) participants, a federal United States food and nutrition program for low-income individuals to increase food security. The studies found that SNAP participants more likely to shop at larger stores [10] that are not necessarily the stores closest to where they live, and that SNAP may mitigate high cost-of-living expenses and assist with improving diet among resource-constrained households [9]. To date, however, no studies have examined the relationship between food prices at one’s main food retail venue for shopping, distance to one’s main food retail venue for shopping, and food security among residents of a geographic area that transitioned from being a food desert to having a full-service supermarket, resulting in improved access to healthy foods.

This analysis examined the possible role of food-store prices and the distance to where study participants reported doing their major food shopping on food security during a unique time-period when their neighborhood of residence changed from a food desert with low access to healthy foods to one with a full-service supermarket. We used data collected by the Pittsburgh Hill/Homewood Research on Eating, Shopping, and Health (PHRESH) study, limited to residents of the neighborhood that, in 2013, acquired the new full-service supermarket [11]. Overall, study participants who were residents of the changing neighborhood experienced food security improvements between 2011 and 2014 [12,13]. Yet, the reasons why and how food security improved among neighborhood residents are not well understood. We set out to evaluate whether place-based and/or price-based factors were associated with the food security improvement within this changing neighborhood.

In this analysis, we mirrored a previous analysis where members of our team examined the relationship between distance to where residents did their major food shopping, food prices at the retail venue, and obesity [14]. Here, we examined food security as the dependent variable to help answer if place and/or price-based policy initiatives may be associated with improved food security for low-income food desert residents [11]. We hypothesized that there would be a statistically significant association between both the distance to the major food shopping store as well as food prices at the major food shopping store with an individual’s level of food security.

## 2. Materials and Methods

In 2011, a random sample of households (*n* = 1372) from two low-income, Pittsburgh neighborhoods were enrolled into PHRESH [11]. The primary adult food shopper from each of the enrolled households completed in-person interviews which collected information including food shopping behaviors, food security, and dietary intake. This analysis focuses on residents in one of the study neighborhoods—the Hill District—which underwent a transition from a food desert neighborhood to one with a full-service supermarket. The research team followed the same households (*n* = 897) in each successive survey wave and in the case where participants had moved from their address, recruited new members into the study. The analysis sample only includes study participants with complete data. We describe how we arrive at the analysis sample in detail below.

Baseline data were collected in 2011 (prior to the supermarket opening in 2013), with two follow-up waves of data collection of the same households in 2014 and again in 2018. At each wave, an in-person 80-min survey was administered which collected study participant’s age, gender, race/ethnicity, education, income, marital status, number of children in the household, and access to a car. Survey participants were asked for their home address and the name and location of their major food shopping store for street network distance measures. “Major food shopping store” was identified by asking, “What is the name and address of the main store where you most often do your major food shopping?”.

Participants were also asked whether they or any member of the household received SNAP in the last 12 months. Those who responded affirmatively in all waves that they were surveyed in were categorized as “continuous” SNAP participants [12]. Food Security was measured using the USDA Adult Food Security Survey Module [15] with higher scores indicating greater food insecurity. Improved food security was defined as reduction in the continuous food insecurity score by two points or more between the 2011 and the most recent assessment of the study participant either in 2014 or 2018.

At each wave (2011, 2014 and 2018), trained data collectors also collected food prices from audits of all stores that sold food and were located within the geographic boundaries of the neighborhood, as well as the top ten most frequented stores participants reported going to for their major food shopping outside the neighborhood in each survey wave. Adapting methodology established using the Bridging the Gap Food Store Observation Form, a price index for each store was created [16]. The standardized price index was a z-score based on the prices collected on prices for food staples, junk foods, fruits and vegetables. The store-level standardized price index was calculated by subtracting the mean and dividing by the standard deviation for the food staples, junk foods, and fruits and vegetables indexes and then summing them. The price index for food staples was calculated based on twelve eggs, half gallon (1.89 L) of whole milk, 567-gram loaf of white bread, 425–510-gram box of high-sugar cereal and 425–510-gram box of low-sugar cereal. The price index for junk food is calculated using the sum of the least expensive soda unit multiplied by 2000 g for a two-liter family-size bottle of soda as well as the least expensive unit price of a 312-gram bag of chips. Finally, the price per kilogram for six items was used to calculate the fruit and vegetable price index: apples, bananas, lettuce, oranges, potatoes and tomatoes [14].

For this analysis, we merged survey responses at each wave with the GIS-calculated distance between the participant’s home and their reported major food-shopping store, as well as the store price data collected by audits from each store. Distance to the food retail venue from each study participant’s home in each wave was computed by geocoding both home and store addresses using ArcGIS. The distance between the two points was calculated using the shortest route along an existing road network [14]. Informed consent was obtained from all subjects involved in the study. It is important to note that all of the measures used in this study were asked and defined consistently across survey wave. All study protocol was approved by the RAND Corporation’s Human Subjects Protection Committee.

We performed two analyses. The first analysis examined characteristics that were associated with (within-person) improvements in food security. For this analysis, the sample included panel respondents who lived in the Hill District in 2011, with at least two food insecurity measures, a baseline (2011) food insecurity measure, and complete sociodemographic and store-related data. The analytic sample was comprised of 518 study participants who provided at least two food insecurity measures. We compared characteristics of study participants whose food security improved with those whose food security did not improve between their first and last wave of data collection and used chi-squared tests and t-tests to detect significant associations.

The second analysis examined demographics, food price index, year of survey, and distance to major food-shopping store as predictors of food security, using repeated cross-sectional linear regressions. The pooled analytic sample was comprised of 1738 observations across all three waves. For each wave, the sample included respondents who lived in the Hill District with complete information across demographics, distance to their major shopping and major shopping store price information. At baseline, 762 study participants met these criteria. We excluded 135 respondents who were missing one or more values for predictors described above. In 2014 and 2018, 78 and 53 participants were excluded due to missing data, out of 599 and 508 total participants, respectively. To assess the effect of excluding individuals with missing values, we compared the baseline characteristics of participants in the analytic sample to those who were excluded. The two groups were similar except that those excluded were more likely to have children in the household (see Appendix A
Table A1). All analyses were conducted with the sample of individuals without missing data in any of the variables.

For the second analysis and in order to improve statistical power and precision, we pooled three waves of data for the estimation of cross-sectional linear regressions with the food security scale as the dependent variable including fixed effects for year to account for the effect of time. In the first regression model, we included sociodemographic characteristics and SNAP participation status as covariates (Model 1). Then, we added food price index associated with the participant’s major food-shopping store (Model 2), distance to major food-shopping store (Model 3), and both the food price index and the distance to major food-shopping store (Model 4), in addition to the characteristics included in Model 1. All analyses were conducted in 2020 in SAS software version 9.4.

## 3. Results

Table 1 shows baseline descriptive statistics of the study participants in the analytic sample. About half of study participants were 55 years or older, 79% were female, 95% were Black, and 44% had an education level beyond high school. About half of study participants reported per capita annual household income below $10,000, 15% were married or living with partner, and 26% had children in the household. Finally, 51% of study participants either owned or had a car they could borrow when needed and 35% reported continuously participating in SNAP in the past 12 months. On average study participants had a food security score of 2.1 (SD = 2.6). The average distance to the major shopping store was approximately 3.4 miles (SD = 3.0), the average standardized price index for the major store was −0.4 (SD = 1.0), and the average staple price at the major shopping store was 12.5 (SD = 0.7).

In Table 2 (columns 1 and 2), we compared characteristics of study participants experiencing improvement in food security between 2011 and follow up in either 2014 or 2018 (30%) to those of households without improvement in food security (70%). Improvement in food security (*p* < 0.05) was found among individuals with education less than high school, with less than $10,000 in annual household income per capita, and continuous SNAP participants. These unadjusted (or, bivariate) analyses show that study participants with a lower socioeconomic status exhibited gains in food security. In these unadjusted (or, bivariate) analyses, we did not see associations between improved food security and age, race/ethnicity, gender, marital status, child in the household, car ownership, distance to major food shopping store, or by food price index of store. The results imply that sociodemographic characteristics related to education and income could be important predictors of improvements in food security.

Table 3 shows the results for each of the regression models with continuously measured food security as the outcome. Positive regression coefficients imply an increase in the USDA Adult Food Security Survey Module score, indicating worse food security. Negative regression coefficients signify declines in the USDA Adult Food Security Survey Module and can be interpreted as better food security. In Model 1, the following measures indicated higher food security: per capita household income, having a child, and access to a car. We also observed higher food security in each of the follow-up survey waves relative to baseline, as exhibited by a negative coefficient for each wave measure. The latter result implies that food security improved in the Hill District over time. Model 2 illustrated no association between the food prices and food security, and the magnitude and statistical significance for the other covariates were comparable to those found in Model 1. However, Model 3 showed that increased distance to where residents reported traveling for their major food shopping was associated with higher food security. In other words, individuals who travelled further for their food shopping had higher food security. Results for the other covariates for Model 3 were similar to those in Models 1 and 2. The results show that study participants with a higher household income, resided in a household with kids, and/or who had access to a vehicle exhibited increases in food security. In Model 4, which was fully adjusted with all covariates, we observed that only distance to the major food shopping store was significantly associated with better food security. In other words, in a model adjusting for age, sex, education, household income, children in the household, marital status, and vehicle access, the farther an individual traveled for food shopping, the more food secure they were.

## 4. Discussion

There continues to be a robust conversation on the role of the local food environment in health and nutrition outcomes [17]. Although there is a complex relationship between geography and nutrition, we know that residents of food deserts are at higher risk of food insecurity, and this is especially true for low-income, racial and ethnic minority populations [18]. This analysis compared residents of a food desert that transformed into a neighborhood with a full-service supermarket comparing those whose food security improved to residents whose food security did not improve. We additionally examined whether distance to residents’ major food shopping store as well as food prices were associated with food security. Our analysis included several years of survey data and found that distance travelled to main food shopping was positively associated with greater food security. This association was statistically significant even after controlling for access to a car. We found that food prices of where one shops was not associated with food insecurity. This result is in contrast to other existing studies [7]. One possible reason for our study having a different result is that the sample is exclusively low-income adults living in a former food desert.

Our results suggest that closer proximity to one’s major food retailer may not necessarily be associated with better food security. One possible explanation for this finding is that residents who traveled further had the capacity in ways we were unable to control for which may have translated into improved food security, or fewer hurdles with access to sufficient food. Another possibility is that the store types that participants travelled to which were further (e.g., big box stores) allowed individuals to purchase increased quantities of food in bulk, which protected households against food insecurity.

We found higher rates of food security for families with a child in the household. This is a surprising result given it is in contrast to national statistics that indicate households with children are less food secure [19]. It is also important to note that we found that better resourced households exhibited improvements in food security. Specifically, higher per capita household income and having access to a car led to increases in food security. This result is consistent with the existing literature [20,21]. The result is notable given that the study sample resides in a low-income neighborhood. Future research should explore possible mechanisms for the differences in food security based on household income and having access to a car, as well as why households with children in our study sample are more food secure.

Some of our published prior findings suggest that changes in the food environment have the potential to impact food security [12]. But the exact mechanism for the improvement was largely unexplored. Our findings are consistent with a diverse array of studies that have found that people are willing and prefer to travel outside their own neighborhood for food shopping based on price and their individual taste preferences [10,22,23]. For example, LeDoux and Vojnovic found that residents of a disadvantaged food dessert in Detroit, Michigan were more likely to shop at independent, discount, and regional supermarkets located outside of their neighborhood [24]. In a second study of the same Detroit community, they also found that residents of the disadvantaged neighborhood were more likely to travel further, if necessary, to purchase fruits and vegetables [25]. These two studies in Detroit, combined with the present study in Pittsburgh PA, demonstrate that individuals may travel further to purchase healthy foods which could potentially increase food security. Given that our study finds an association between distance to major food shopping store and food security, more work is needed to understand and unpack this finding to identify underlying causal mechanisms.

## 5. Limitations

This study has several limitations. First, our results are limited to a neighborhood that underwent a transformation from a food desert to one with a full-service supermarket, but still may not be generalizable to all former food deserts. Second, we use distance and price metrics of the stores where participants reported doing their major food shopping; we do not include measures for additional food stores or measures of travel time. Third, we do not have data on the specific items that were purchased by the study participant. Finally, our results should not be considered causal. Instead, we highlight possible correlates of food security among predominantly low-income resides of a former food desert.

## 6. Conclusions

While proximity to a full-service supermarket is associated with food security, travelling further for one’s major food shopping was associated with higher food security. Public health officials and policymakers should continue to examine where residents acquire their food and acknowledge that the associations between proximity to full-service supermarkets, actual distance residents travel for major food shopping, and individual food security are complex and nuanced.

## Figures and Tables

**Table 1 nutrients-14-05242-t001:** Characteristics of the analytic sample in the Hill District.

	Baseline Characteristics (*n* = 518)	
Age Group	*n* or mean	% or SD
Age 18–34	71	13.7
Age 35–54	177	34.2
Age 55–74	209	40.3
Age 75+	61	11.8
Gender		
Male	109	21.0
Female	409	79.0
Race Category		
Black	493	95.2
Other	25	4.8
Education Category		
Less than high school	71	13.7
High school degree	216	41.7
Some college	166	32.0
College or graduate degree	65	12.5
Per Capita Household Income		
$0–$4999	83	16.0
$5000–$9999	187	36.1
$10,000–$19,999	151	29.2
$20,000+	97	18.7
Marital Status		
Married/Living with partner	77	14.9
Never Married	228	44.0
Widowed/Divorced/Separated	213	41.1
Kids in Household		
Household with kids	129	24.9
No Kids	389	75.1
Car ownership		
Household with access	266	51.4
No Vehicle	252	48.6
On SNAP during all survey waves		
Never/Sometimes used SNAP	340	65.6
Used SNAP (every time)	178	34.4
Baseline major shopping store distance		
Average distance from home to major store	3.4	3.0
Baseline Food Insecurity		
Average food insecurity	2.1	2.6
Price index for major shopping store		
Standardized Price Index for major store	−0.4	1.0
Average Staple Price for major store	12.5	0.7

Note: Individual-level characteristics from the initial year that the study participant was surveyed (2011 or 2014).

**Table 2 nutrients-14-05242-t002:** Descriptive statistics comparing those who did and did not improve food security among Hill District residents.

	Food Security did not Improve between First and Last Wave Respondent Was Surveyed (*n* = 363)		Food Security Improved between First and Last Wave Respondent Was Surveyed (*n* = 155)	
	*n* or mean	% or SD	*n* or mean	% or SD
Age Group				
Age 18–34	45	12.4	26	16.8
Age 35–54	118	32.5	59	38.1
Age 55–74	150	41.3	59	38.1
Age 75+	50	13.8	11	7.1
Gender				
Male	78	21.5	31	20.0
Female	285	78.5	124	80.0
Race Category				
Black	342	94.2	151	97.4
Other	21	5.8	4	2.6
Education Category				
Less than high school	40	11.0 *	31	20.0 *
High school degree	153	42.1 *	63	40.6 *
Some college	119	32.8 *	47	30.3 *
College or graduate degree	51	14.0 *	14	9.0 *
Per Capita Household Income				
$0–$4999	52	14.3 **	31	20.0 **
$5000–$9999	118	32.5 **	69	44.5 **
$10,000–$19,999	110	30.3 **	41	26.5 **
$20,000+	83	22.9 **	14	9.0 **
Marital Status				
Married/Living with partner	58	16.0	19	12.3
Never Married	149	41.0	79	51.0
Widowed/Divorced/Separated	156	43.0	57	36.8
Kids in Household				
Household with kids	83	22.9	46	29.7
No Kids	280	77.1	109	70.3
Car ownership				
Household with access	194	53.4	72	46.5
No Vehicle	169	46.6	83	53.5
On SNAP during all survey waves				
Never/Sometimes used SNAP	255	70.2 **	85	54.8 **
Used SNAP (every time)	108	29.8 **	70	45.2 **
Baseline major shopping store distance				
Average distance from home to major store	3.4	3.0	3.6	3.3
Price index for major shopping store				
Standardized Price Index for major store	−0.3	1.0	−0.5	1.0
Average Staple Price for major store	12.5	0.8	12.4	0.6

Note: Statistical testing between groups was done using a chi-squared test and t-tests, ** *p* < 0.01 level, * *p* < 0.05. Food security is from the USDA’s Adult Food Security Survey Module; lower scores imply greater food security. Improved food security was defined as reduction in the continuous food insecurity score by two points or more between the 2011 and the most recent assessment in 2014 or 2018. Food security did not improve was defined as an increase, no change or staying the same, or decrease of less than two points in the continuous food insecurity score.

**Table 3 nutrients-14-05242-t003:** Linear regression results for association of distance and price with Food Security scores (*n* = 1738).

	Model 1: Participant Characteristics and Wave	Model 2: Includes Price	Model 3: Includes Distance to Store	Model 4: Includes Price and Distance to Main Shopping Store
Distance to main shopping store in survey wave			−0.04 (0.02) *	−0.05 (0.02) *
Standardized price index in main shopping store survey wave		0.04 (0.05)		−0.05 (0.06)
Age				
Age	0.04 (0.02)	0.03 (0.02)	0.03 (0.02)	0.03 (0.02)
Age Squared	−0.00 (0.00) **	−0.00 (0.00) **	−0.00 (0.00) **	−0.00 (0.00) **
Male	0.00 (0.14)	−0.00 (0.13)	−0.02 (0.13)	−0.01 (0.13)
Education				
High School	−0.18 (0.17)	−0.18 (0.17)	−0.18 (0.17)	−0.18 (0.17)
Some College	−0.08 (0.18)	−0.08 (0.18)	−0.08 (0.18)	−0.08 (0.18)
Bachelors degree or higher	−0.13 (0.22)	−0.13 (0.22)	−0.14 (0.22)	−0.13 (0.22)
Per Capita Household Income 000s	−0.03 (0.00) **	−0.03 (0.00) **	−0.03 (0.00) **	−0.03 (0.00) **
Household with kids	−0.74 (0.16) **	−0.74 (0.16) **	−0.73 (0.16) **	−0.73 (0.16) **
Marital Status				
Married/Living with Partner	−0.23 (0.17)	−0.23 (0.17)	−0.22 (0.17)	−0.22 (0.17)
Widowed/Divorced/Separated	0.30 (0.14) *	0.30 (0.14) *	0.30 (0.14) *	0.29 (0.14) *
Access to Vehicle	−0.44 (0.12) **	−0.43 (0.12) **	−0.41 (0.12) **	−0.41 (0.12) **
Survey Wave				
2014	−0.74 (0.13) **	−0.72 (0.13) **	−0.75 (0.13) **	−0.77 (0.13) **
2018	−0.61 (0.14) **	−0.57 (0.15) **	−0.64 (0.14) **	−0.69 (0.16) **
SNAP participant	0.16 (0.12)	0.16 (0.12)	0.16 (0.12)	0.15 (0.12)
Intercept	3.17 (0.54) **	3.20 (0.55) **	3.35 (0.55) **	3.35 (0.55) **

Note: Regression model used was ordinary least squares. Outcome measure is linear measure of food security. Food security is measuring using the validated 10-item USDA Adult Food Security Survey Module. A higher number on the food security scale (0–10) indicates greater food insecurity. Standard errors are contained in the parentheses. ** indicates *p* < 0.01 level, * indicates *p* < 0.05.

## Data Availability

The data presented in this study are available on request from the corresponding author.

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
