# Peer review of "What Is Associated with Changes in Food Security among Low-Income Residents of a Former Food Desert?"

_nutrients, 2022, doi:10.3390/nu14245242_

Round 1
Reviewer 1 Report
The subject of the study is very interesting, current and necessary. However, the study methodology is not clear, mainly referring to the sample and study participants. This does not allow for an adequate scientific analysis of the manuscript. It is not clear whether the same families were followed in all the years of study. Apparently, there are different samples for each wave. However, the study does not assess whether these distinct samples can be compared or pooled. Without these details, it is impossible to assess whether the observed results are due to selection bias.
Author Response
Reviewer 1
The subject of the study is very interesting, current and necessary. However, the study methodology is not clear, mainly referring to the sample and study participants. This does not allow for an adequate scientific analysis of the manuscript. It is not clear whether the same families were followed in all the years of study. Apparently, there are different samples for each wave. However, the study does not assess whether these distinct samples can be compared or pooled. Without these details, it is impossible to assess whether the observed results are due to selection bias.
Response: We thank the reviewer for their positive feedback. The reviewer is correct that the study’s data collection utilizes a panel design; that is, the same households and their participants were followed over the course of the study. Our regression analysis pooled participants in each survey wave, adjusting for year, to improve power and precision. Any differences in sample composition will stem from nonresponse within a survey wave – as a result, we have included an analysis below to demonstrate that those with any missing data are similar to those without missing data, with respect to individual characteristics. In the Methods section, we have clarified that we examined the same households over time when looking at improvements in food security over time.
We also report the sociodemographic characteristics for the study participants with no missing data (included in the study), and those with missing data (not included in the study) below (and as Appendix 1). We note that the distributions are similar to one another. The only exception is a statistically significant difference in the variable reflecting having kids in the household. The analytic sample had fewer individuals with a child in the household compared to the excluded sample.
Reviewer 2 Report
Dear Authors,
In connection with the submitted article "What is Associated with Changes in Food Security among Low-income Residents of a former Food Desert?", the following findings are made.
Generally, I thing you should reorganize all sections of manuscript.
Introduction
*The Introduction section should be significantly expanded. I recommend a separate "Literature review" paragraph after the introduction paragraph in order to support the importance of the work. However, the new "Literature review" paragraph should be expanded with more references to other similar papers and research.
*The introduction section should specify what food security is. You are providing the obvious facts that food security is related to the price of food and its availability. Apart from the economic and physical aspect of food security, the authors should pay attention to the health aspect.
*In line 38 you should give more details about Program SNAP What was the program about? Who were the participants in this program?
*The research hypothesis should be formulated.
Materials and Methods
* Due to the multi-stage nature of the research it is recommended to present the subsequent stages of research in a graphic form, with details about each of them (for example number of households). On this figure should be clearly indicate which waves were compared.
* line 68-71: please, give the information about numer of households from one of the study neighborhoods. In line 65 you mentioned about numerr of households (n=1,372), but it were households from two low-income neighborhoods.
* line 72-73: In two follow-up waves in 2014 and again in 2018 it were the same random sample of households (n=1,372) that in 2011?
* Specify, if the survey from 2011 and in two follow-up waves in 2014 and again in 2018 were the same?
* line 92-92: please, give more details abous z-score
Results
* Why the characteristics of sample from 3 waves of research (separatelly) weren't shown in Table 1?
* line 122: in table 2 (columns 2 and 3); In column 1 is "Measure";
* food security measurement results are mising. It was mentioned in the methodology section that it was measured using the USDA Adult Food Security Survey Module [13] with higher scores indicating greater food insecurity. Where are these scores? In Table 2 are presented the percentage of respondents .
Discussion
* The Discussion should be carried out in a broader context. In Model 1 were included sociodemographic characteristics and SNAP participation status.
* Were there any “Limitations of the study”?
Reviewer 3 Report
Food insecurity is linked to poor nutrition, obesity, and chronic disease. The manuscript, the authors mirrored a previous analysis where members of our team examined the relationship between distance to where residents did their major food shopping, food prices at the retail venue, and obesity. They examined food security as the dependent variable to help answer if place and/or price-based policy initiatives may be associated with improved food security for low-income food desert residents. The results showed that proximity to a full-service supermarket is associated with food security, travelling further for one’s major food shopping was associated with higher food security. And the information provided by this research would helpful for public health officials and policymakers to examine where residents acquire their food and acknowledge that the associations between proximity to full-service supermarkets, actual distance residents travel for major food shopping, and individual food security are complex and nuanced. Specific comments are as following.
1. Avoid repetition between the Title and the keywords.
2. Tables 1 and 2 should be three-lined tables.
3. I encourage the authors to present the data in Table 3 in a more visual form.
Author Response
Reviewer 3
Food insecurity is linked to poor nutrition, obesity, and chronic disease. The manuscript, the authors mirrored a previous analysis where members of our team examined the relationship between distance to where residents did their major food shopping, food prices at the retail venue, and obesity. They examined food security as the dependent variable to help answer if place and/or price-based policy initiatives may be associated with improved food security for low-income food desert residents. The results showed that proximity to a full-service supermarket is associated with food security, travelling further for one’s major food shopping was associated with higher food security. And the information provided by this research would helpful for public health officials and policymakers to examine where residents acquire their food and acknowledge that the associations between proximity to full-service supermarkets, actual distance residents travel for major food shopping, and individual food security are complex and nuanced. Specific comments are as following.
Response: We thank the reviewer for their comments and positive feedback.
- Avoid repetition between the Title and the keywords.
Response: We have revised the key words to remove one of the terms that included the word “food” (“Food Desert”) and have included “Nutrition” instead.
- Tables 1 and 2 should be three-lined tables.
Response: We have revised Tables 1 and 2 to be three-lined tables.
- I encourage the authors to present the data in Table 3 in a more visual form.
Response: While we appreciate the reviewer’s comment, we prefer to report the regression results in full so the reader can see each of the effect sizes and their association with the change on food security.

Round 2
Reviewer 2 Report
Dear Authors, there are some major issues needed to be improved still:
*The necessary information about sample should be supplemented; in line 76-78 you wrote that "The research team followed the same households (n=897) in each successive survey wave and in the case where participants had moved from their address, recruited new members into the study"
Table from your coverlatter with demographic information at each waves shows that in the first wave it was 762, in the second - 521, and in the last - 455. Where do these differences come from?
*In line 101- 110 acordind to MDPI Instructions for Authors SI Units (International System of Units) should be used. Imperial, US customary and other units should be converted to SI units whenever possible.
* Due to the multi-stage nature of the research I still recommend to present the subsequent stages of research in a graphic form, with details about each of them.
* Why did you not take into account the professional activity of the the primary adult food shopper from each of the enrolled households?
*Results are presented and discussed in a too succinct manner.
